# A Study on Design and Case Analysis of Virtual Reality Contents Developer Training based on Industrial Requirements

**HoSeong Kang** and **JungYoon Kim** *

Graduate School of Game, Gachon University, Seongnam-Si 21936, Korea; sos6884@naver.com
* Correspondence: kjyoon@gachon.ac.kr; Tel.: +82-31-750-8666

**Abstract:** The fourth industrial revolution has evolved at an exponential pace that has raised the level of virtual reality (VR) technology industry development. The VR-based contents industry has significantly grown in convergence with other areas; however, its production suffers from lack of skilled human resources. In this regard, this paper provides a study on educational courses dealing with VR contents production. The study identifies the current status of VR contents industry and evaluates the training contents currently being used at relevant training institutions. As a result, an industrial demand-customized educational model and its operations based on cooperative relationship between training institutions and industrial companies by means of cooperative projects and mentoring was designed. The outcome of the evaluation on the results of training courses operation served as a basis for the design of the proposed educational model and indicates the effectiveness of the training.

**Keywords:** virtual reality (VR) content; education model; education operating model; human resource training

## 1. Introduction

According to "The Top 10 Strategic Technology Trends for 2019" released by Gartner, an international research organization at Gartner Symposium & ITxpo, VR technology was selected as one of the technologies that would change the world [1]. Contents allowing people to experience VR have been massively increasing in conjunction with VR technology evolution which started from the study on head mounted display (HMD) by a computer scientist Ivan Edward Sutherland in 1968 [2,3].

In addition, Steven M. Lavalle stated that VR technology is rapidly evolving, which makes it undesirable to define VR [4]. Moreover, other scientists generally define VR technology as "a technology that allows people to build virtual world within computer space using people's imagination, interaction using five human senses such as seeing, hearing and touching, and indirectly experience some situations that are likely very rare to happen in the real world" [5,6].

Furthermore, higher levels of flow and presence in such experiences can be felt through VR contents as compared to non-VR contents [4,7]. VR technology easily replaces situations in various fields that might be difficult to encounter in real life and allow people to experience situations with a higher level of flow and presence in an indirect manner which minimizes cost and time. For these reasons, VR contents are now actively created and used as alternative means in diversified fields such as leisure, education, military, and medication, as shown in Table 1 [8,9].

**Table 1.** Virtual Reality (VR) contents industry trends.

| Field | Company | Description of Contents |
|-------|---------|-------------------------|
| Military + Education | FRONTIS | VR-based educational contents on maintenance theory/practice in association with military maintenance support system [10,11] |
| Military + Medical | USC Mixed Reality Lab, ICT | VR contents designed to treat military soldiers who have suffered from post-traumatic stress disorder [12] |
| Webcomics | Dexterstudios | Interactive manga contents combining traditional webcomics and VR [13,14] |
| Education + Medical | Discovery Lab | Educational VR contents designed for medical field supporting anatomical body [15,16] |
| Social | VRCHAT | VR social contents which allows users to easily design in VR space [17] |
| Construction | InsiteVR | Cooperation and communication contents in VR environment using construction design diagram and model [18,19] |
| Medical | VIVID vision | Eye movement assisting contents for treating eyeball and vision using VR [20,21] |
| Vehicle + Entertainment | AUDI | VR entertainment contents connecting with movement of vehicle in real time basis [22] |

The VR contents industry unfortunately has a limited number of qualified human resources. The limited number of VR personnel emerged as a problem while a massive amount of money from Korean government and investors flow into the VR industry, and numerous small/medium sized and startup companies proceed with projects involving VR contents production.

In this regard, there is an increasing demand for skilled VR human resources. VR contents production require highly skilled developers who have a profound and comprehensive understanding on new techniques and rapidly changing technology. In addition, VR contents need to be designed and developed to avoid sickness and dizziness that might be caused by cognitive dissonance while maintaining a reasonable level of flow and presence [23].

The Software Policy & Research Institute (SPRI) in South Korea forecasted that the demand of AR/VR contents production-related jobs, which include designers and developers, will increase in the future. In 2018, the demand for AR/VR software workers had reached 483. Considering the growth rate of the relevant market, the demand of software workers is predicted to reach 19,847 in 2022 [24].

Various efforts in different ways have been made by the Korean government and training organizations in order to meet this increasing demand of VR contents production human resources. They have invested money that involves the operation of training courses focused on fostering human resources fit for the industrial requirement. However, in spite of these efforts, the supply of specialized workforce for VR contents market is still insufficient. Thus, this paper aims to design effective training courses focused on fostering and training skilled people that can satisfy the industrial needs with the support from Korean government in order to address these issues.

## 2. Related Research

As an effort to foster skilled human resources involving VR contents, related curricula were implemented by universities. In addition, training courses were provided as part of government projects, as shown in Table 2.

The curriculums offered by these educational institutions are primarily being carried out in two steps: (a) basic knowledge learning, and (b) project-based practice. They have applied a problem-based learning (PBL) approach that is commonly used in fields of modern engineering and software training. Problem-based learning theory was formulated to address the social issues that emerged from graduates of engineering universities.

The educational institutions in Table 2 conduct theoretical curricula. On the other hand, the education method proposed in this paper has the distinction of cooperating with industry through industrial projects.

The qualification of graduates has a significant gap with industry needs. The graduates lack the capability to create, design, integrate, and communicate. In this regard, a problem-based learning model was implemented in the college of engineering in order to address this social issue [24].

**Table 2.** Educational institutions and courses related to VR content.

| Name of Educational Institution | Department Where Training Is Provided | Name of Course or Degree | Description |
|---|---|---|---|
| Youngsan University | Undergraduate school | Major in Virtual reality contents | This major covers the overall range of VR contents production. Students will learn how to combine VR with traditional contents. Students are expected to become specialists in contents production. |
| Daekyeung University | Undergraduate school | Major in VR/AR | In this major, students will learn the overall techniques necessary for producing VR contents so that they will become specialists in practical sense. |
| Hongik University | Graduate school | Major in VR/AR contents, Graduate School of Film and Digital Media | This major provides training courses that combine arts and technology. It aims to foster practical experts in creating VR/AR contents and are capable of planning, developing and training MR (VR/AR) contents. |
| Chung-Ang University | Graduate school | Major in CG/VR, Graduate School of Advanced Imaging Science | This major aims to train and cultivate people who can use new presentation techniques, develop new environments and cutting-edge applications through studying various topics such as 2D, 3D video, modeling, rendering, and animation. |
| Gyeonggi Content Agency | Government subsidized training course | Basic course in VR/AR Academy | It is a government subsidized course designed to cover VR/AR contents/game and 360 degree video production. The course aims to develop VR/AR specialists. |
| Korea Radio Promotion Association | Government subsidized training course | 3D VR/AR contents creation course using Unity | This course provides trainings focused on game/non-game VR/AR contents production. |
| Korea Chamber of Commerce and Industry | Government subsidized training course | VR application software developer course | In this course, students can learn how to design, implement, and test software functions for VR services using computer programming language. |
| SVVR | Private training institution | VR content developer course using Unity2017 | It is a basic level of training which teaches how to develop VR contents running on HMD like Oculus and Gear VR using Unity. |

Various studies have been conducted to evaluate the effectiveness of problem-based learning in software and engineering education. The traditional theories based on case studies involving problem and project-based learning were studied by Kim Moon-Soo (2015) to seek the desired direction towards the improvement of domestic education in the engineering fields. He has proposed that a multidisciplinary training approach was necessary with the support of universities and professors for the purpose of improving student's skills to become capable of dealing with practical problems that are likely to happen in the industrial fields [25].

Lee Keun-Soo (2015) conducted a study focusing on problem-based learning in a specific introductory course of computer science in the college of engineering. The results of his study determined that problem-based learning is an essential approach in obtaining expertise which is a critical quality to becoming a competent engineer in modern industrial field [26].

On the other hand, problem-based learning has also been implemented in gaming field which is closely related to VR contents. Lee Dong-Eun (2018) conducted a study on problem-based learning education model mainly intended for game development. He asserted that problem-based education model could effectively provide a learning environment in which students are able to subjectively organize their learning. He also suggested that trainings will become more effective if it will be designed in association with the industrial partners by establishing a cooperative relationship with a specific company at the start of the curriculum design in pursuit of meeting the company's requirements [27].

Problem-based learning (PBL) is an effective way of learning in academic area of software and engineering field. To take advantage of this approach, this study aims to apply PBL on designing a training course for VR contents development. Considering that VR contents industry is consistently changing and evolving by nature, the traditional PBL approach should be adapted to fit the characteristics of VR contents development when designing the education model. There are training institutions that already runs training courses for the emerging VR contents development. However, in the existing training courses, it would be difficult to learn practical knowledge that might be necessary in the actual field since the courses were designed to focus on learning the general theories and skills regarding contents development engine.

Every industry needs human resources. However, they do not want to hire people with no hands-on knowledge and skills for real-world applications. Internship programs were implemented by some companies in order to bridge this gap. Unfortunately, it is not easy for small/medium sized or start-up companies due to insufficient funding and time. This issue gets seriously worse especially in South Korea because the current size of VR contents industry and the number of related companies were too small to supply sufficient human resources.

VR industry is currently growing so many companies were having a difficulty in finding skilled and specialized personnel. Thus, it is necessary to design and operate training courses in conjunction with solid cooperative relationship between training institutions and industrial companies in order to address this issues.

VR technology has been increasingly applied to various fields aside from gaming such as medical, military, and transportation as the number of R&D projects is also growing. In addition, the importance of specialized personnel in VR contents development gets more significant. However, the researches and studies dealing with enriching human resources that deals with VR contents production were not sufficiently conducted. Although there were universities which implements VR contents-related curriculum for undergraduates, VR contents training needs to be incorporated as most of their curriculum were limited to traditional topics like software and game development.

Considering the current status of VR contents field from academic and industrial perspectives, this study aims to design an industrial requirements-based training course consisting of real-world problems and projects that fits the industrial requirements based on cooperative relationship between educational institution and industrial companies.

## 3. Design of Educational Model to Foster VR Contents Developer Training

### 3.1. Overview of the Educational Model

The purpose of VR contents developer training course is to cultivate skilled human resources who are capable of designing and implementing VR contents on the basis of solid knowledge required by rapidly changing industry and not just staying in general software developer or programmer. The design of the proposed training course utilizes the problem-based learning (PBL) to enrich qualified personnel to cope with the VR industry's needs and extend the supply of qualified human resources [28,29].

As shown in Figure 1, the proposed education model is based on cooperative relationship between educational and industrial parties. The industrial party can complement the hands-on knowledge and skills involving VR contents production which are barely provided by the educational party so that trainees could acquire hands-on knowledge that might be required in the actual operations of the industry. Considering that a project is generally carried out by a team in actual industrial field, the training model is designed to organize a team and fulfill a team-based project so that trainees could learn how to work in a team-based environment and be able to manage team-based projects.

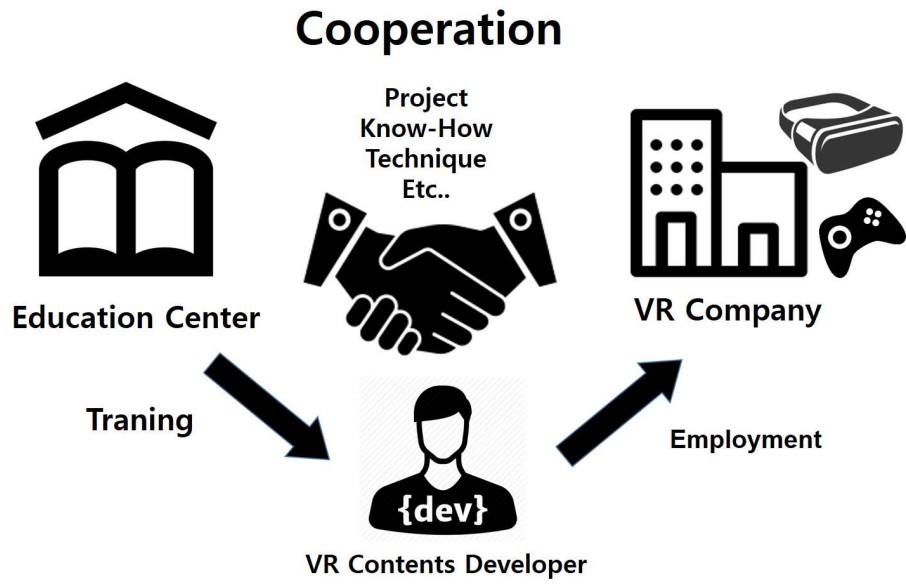

**Figure 1.** VR Contents Developer Educational Model Overview.

The proposed design consists of an educational model design and its operating model design. The educational operating model describes the educational model operations which primarily deals with the cooperation process between the educational and industrial parties through a cooperative linkage. In adopting a PBL approach, the educational model details the training contents in every stages of basic training, advanced training, and project-based training to provide highly trained and qualified personnel in accordance with the industrial requirements.

### 3.2. Education Operating Model for VR Content Development

The step-by-step process for cultivating VR contents development are as follows:

- Step 1. Make an agreement of cooperation with a VR contents industrial party
- Step 2. Review the training contents along with training personnel, industrial party, and lecturers.
- Step 3. Launch training courses and carry out basic and advanced trainings.
- Step 4. Request for an industrial project requirement from a cooperating company (i.e., industrial party).

- Step 5. Proceed with team-based fulfilment of the project requirements and provide mentoring.
- Step 6. Present project outcomes and discuss hiring with the cooperating companies.

In the first step, the training institution enters into an agreement of cooperation with an industrial party. Prior to the selection of cooperating companies, an educational institution has to set a selection criterion that includes the size of company, relativeness in terms of the company's industrial characteristics, and willingness to hire trainees. Among these criteria, the company's willingness to hire is the most critical factor since the primary purpose of the training is to supply qualified personnel into the industry and address the limited human resources issue. This agreement would be essential to propel the cooperation between the two parties and ultimately facilitate mutual prosperity and advancement.

The scope of responsibilities of both parties can be described as follows:

1. The industrial party proposes the training topics suitable for industry-customized projects and provides mentoring to trainees.
2. The educational party develops a curriculum in accordance with the industrial party's requirements and implements it to enrich qualified personnel equipped with specialized skills.
3. Both the educational and industrial parties mutually help each other in the employment for the qualified trainees.

In the second step, the industrial party, educational institution, and assigned lecturers conduct a review of the training contents. The industrial party is allowed to participate in the review process to ensure that the training contents will be in accordance with the industrial requirements. The review can guarantee that the training course has been designed to enrich the trainee's capability as required by industrial needs. In addition, this would be a good opportunity for a training institution to acquire up-to-date information regarding the emerging cutting-edge technologies and trends in the actual industry operations, and come up with a better training curriculum to nurture highly qualified personnel suitable for industry requirements.

In the third step, the training courses were implemented based on the designed educational model. The trainees could strengthen their development capability through undergoing three different levels of training: (1) learning the basic and technical background of VR contents; (2) learning the basic skills necessary for development through practice; and (3) learning advanced skills. Through these learnings, the trainees are expected to be able to carry out an industrial project and participate in developing VR contents. While on training, it is essential to keep interacting with the industrial party and get feedback or opinion to guide the trainee's to become capable for industrial needs [25,30].

The fourth step is the preparation for undergoing an industrial project. The educational institution requests for a project requirement from the industrial party. The industrial party specifies the details of the project which aims to evaluate the trainee's development capability. The educational institution prepares fulfillment of the project as soon as the industrial party sends the project requirements. The requirements document specifies the project details including the information about the company and mentor who will be in charge of the project. The requirements document will be reviewed by the educational institution and evaluates the project's feasibility in terms of duration and available resources such as the number of team members required to fulfill the project. The project details can be adjusted or modified in accordance with the requirements from the industrial party.

Figure 2 shows an example of industrial project requirements.

Ⅰ. Introduction to Company

| Company name | | | |
|---|---|---|---|
| business | | Number of workers | |
| location | | | |

Ⅱ. Mentor Info

| Name | | Department | |
|---|---|---|---|
| H.P | | E−Mail | |
| Mentor Part | | | |
| Mentoring Date | | | |

＊ Mentoring participation dates are prepared once a week and on a four−hour basis (consulting date and time can be discussed)

Ⅲ. Project Introduction and Demands (Plans)

| Project name | |
|---|---|
| Project Introduction | |
| Project Objectives | |
| Project Content | |
| Major functions | |
| Project Period | |
| Other items | |

**Figure 2.** Industrial Project Requirements.

In the fifth step, the trainees are expected to carry out the project as specified in the project requirements document. It is essentially important to establish a network between the cooperating company and trainees. An introductory session can be conducted wherein the overview of the project will be presented and both parties will be introduced. The introductory session allows the trainees to familiarize with the counter-part company and personnel who will work with them and know the overall details for the project. The person in charge of the project from the cooperating company is expected to set a meeting with the trainees once a week to provide mentoring. During this meeting, the progress of the project will be monitored and the person in charge from the cooperating company assists in improving the trainee's problem-solving skills in practical ways. The educational institution and lecturers play significant roles of bridging the linkage between the trainees and the cooperating company. They also serve a supplementary role in teaching specific knowledge towards the project.

In the final step of the training, the project outcomes will be reviewed and reflected. The project outcomes need to be summarized and presented through demonstrations and/or presentations. External experts can be invited for the outcome presentations to enrich the review process and instill their feedback regarding the project outcomes. Then, job offering to trainees and hiring should be discussed by means of the solid network between the trainees and cooperating company which has been built while the project was carried out. At this stage, the educational institution plays a significant role on ensuring that more trainees can be employed by the cooperating companies.

*3.3. Educational Model for VR Contents Developer Training*

The educational model for VR contents developer training is shown in Figure 3. The VR contents developer educational model consists of four different steps: (1) learning the development methodology, (2) learning application, (3) VR development application, and (4) fulfilling an industrial project.

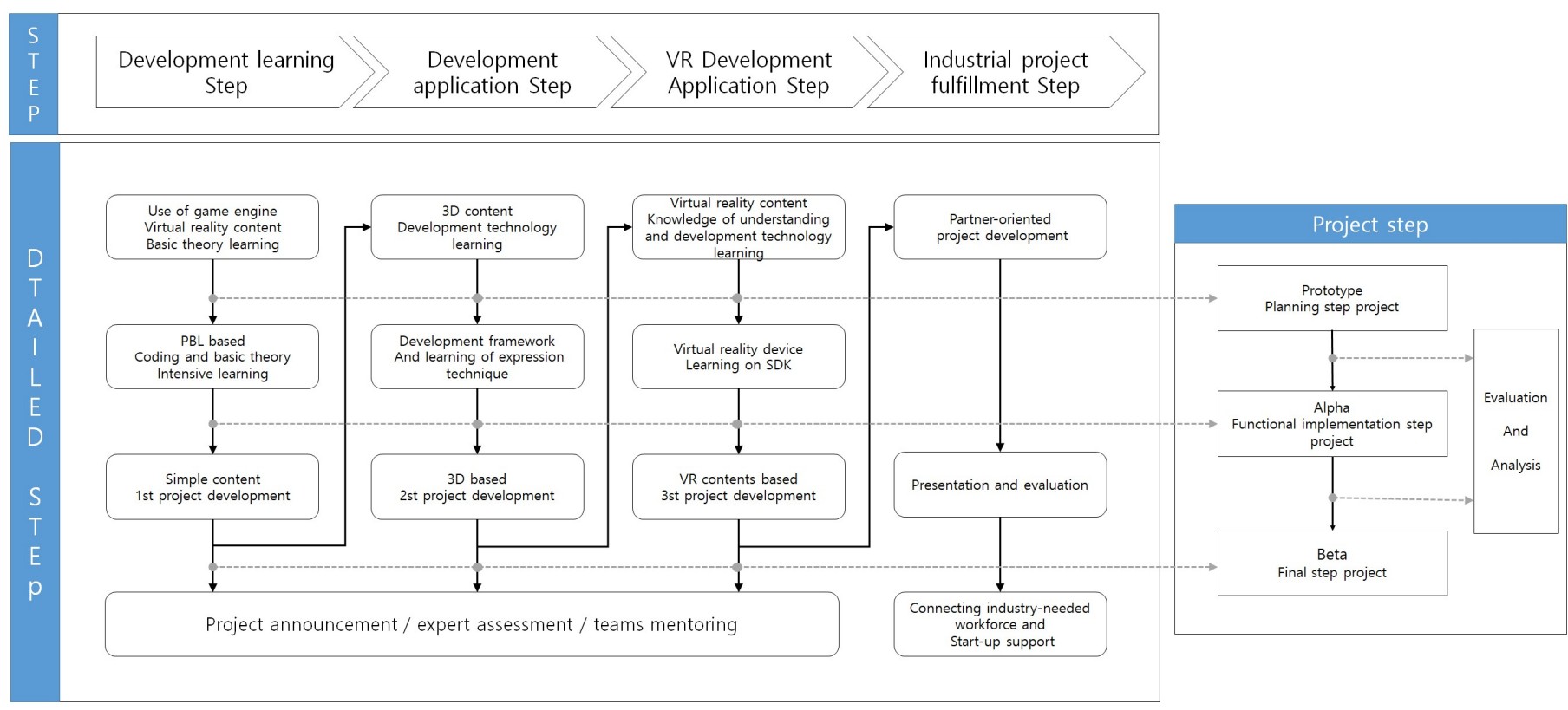

**Figure 3.** VR Contents Developer Educational Model.

First, the trainees will learn the development methodology. They will learn how to use a game engine which is an electronic publishing tool, and be able to understand the basic principles and theories necessary for contents production. The focus of the training in this stage will be on the basic theories since the knowledge level of trainees tends to be diverse in terms of their IT knowledge and development capability. The training contents also focuses on coding and basic understanding in association with problem-solving skills. The trainees are expected to be able to produce small contents with simple logic.

Next, the trainees will learn applications development. The trainees will be able to understand the basic principles and relevant theories in creating VR and 3D contents. They also will learn the contents development framework and acquire advanced development skills. This training contents in this stage was designed to strengthen and refine the trainee's basic skills that has been acquired from the previous basic training. In addition, hands-on skills in using publishing tools will be acquired which will be useful in the actual industrial operations. The trainees are expected to create 3D-based contents after completing this stage of the training.

Then, the trainees will learn the specific techniques in VR applications development. Based on the theories and skills acquired from the previous training stage, the trainees are expected to have practical capabilities and qualifications necessary for developing VR contents. The trainees will be able to identify various VR devices, relevant interfaces, and the use of VR libraries. The trainees are expected to carry out projects on VR-based contents production after completing this stage of training.

Finally, the trainees will carry out an industrial project in cooperation with the counter-part company. Project teams will be organized among the trainees and team members acquire development skills and practical capabilities with the support of the cooperating company. Weekly meetings will be held to provide mentoring to the trainees, thus, they will effectively get vivid experience like working at the actual field. The mentors and lecturers assist in enriching the trainee's qualifications both in theory and development skills.

The step-by-step training approach was designed to improve the communication/presentation skills and enhancing the technological application capabilities necessary for implementing core components. The curriculum was designed to adopt PBL approach as suggested by precedent studies. The lecturers facilitate the learning of theories and technologies to the trainees. Case studies were assigned to the trainees for enhancing their problem-solving skills.

Building the prototype project aims to create an initial project model and making use of the basic theories and skills acquired from the basic training. The alpha version of the project aims to implement the functions of the system model and making use of acquired knowledge and skills inform the advanced training course. The beta version of the project requires the trainees to complete and finalize the project outcomes. The beta version of the project is expected to be a completed form of a system and runnable.

All project versions need to be reviewed and evaluated. The lecturers will be responsible to guide the review process for both the prototype and alpha versions of the project through project outcome assessment and identifying insufficient points. The experts or mentors will be responsible to assess the project outcomes for the beta version of the project wherein they provide feedback and guidance to improve the trainee's project fulfilment capabilities.

The detailed training contents of this educational model can be modified by reflecting the current changes in the actual operations of the industry as a result of the review process done in "step 2 Review the training contents along with training personnel, industrial party, and lecturers" in Section 3.2.

## 4. Results Analysis of Educational Model for Enriching VR Contents Developer Training

### 4.1. Research Subjects

The design of the educational model d was implemented to "software-based industry-customized training course for VR/AR developers". The course was launched in 2018 with the support of the

Institute of Information and Communications Technology Planning and Evaluation (IITP). The course was implemented in six months from September 2018 to February 2019 with 960 training hours for 27 trainees (i.e., 21 males and 6 females/graduates-to-be or unemployed). Industrial experts from VR industries and academic professors from universities were invited to interview and select candidate trainees for the training course

The selection criteria were focused on the likelihood of employment and/or starting their own business like a start-up company. In addition, the candidate's interest in VR technology and the level of understanding software technologies were also considered. The demographic data for the candidate trainees are shown in Table 3.

**Table 3.** Demographic data for the trainees.

| Type of Category | Category | Amount | Ratio |
|---|---|---|---|
| Gender | Male | 21 | 77% |
| | Female | 6 | 23% |
| Age | 20~24 | 7 | 26% |
| | 25~30 | 15 | 56% |
| | 30~40 | 5 | 18% |
| Academic degree | Up to high school | 3 | 11% |
| | Undergraduate | 12 | 44% |
| | Bachelor/Master degree | 8 | 30% |
| | Graduate | 4 | 15% |
| Whether having related major or not (software/content development) | Related major | 19 | 70% |
| | Non-related major | 8 | 30% |

### 4.2. Case-Based Results Analysis

Prior to opening the training courses, an agreement of cooperation will be entered by the educational institution and the industrial party. This study has evaluated the current status of companies around VR industry in 2018 in order to achieve the purpose of cultivating the industrial needs for specialized human resources in conjunction with communication with industrial parties. These results in cooperative agreements with several firms in VR industry.

The educational institution introduced its objectives and points out the necessity of this training courses to provide companies with an opportunity to hire qualified personnel through a mutual agreement. Thus, cooperative agreements with 86 companies that includes start-up to small/medium sized companies were made to get a higher chance of hiring. The agreements were set from August 2018 and ends on February 2019. During this period, various forms of cooperative activities were planned and conducted to strengthen the relationship such as industry advisory meetings, industrial project fulfilment and mentoring.

### 4.3. VR Contents Developer Educational Model Analysis

Prior to the start of the training, discussions were made with the cooperating companies to ensure whether essential contents necessary for practical works in the actual industry operations were included in each educational model. The effectiveness of the training contents was verified based on the actual operations and ideas received from cooperating companies. Table 4 identifies the contents discussed in this process.

**Table 4.** Demographic data for the trainees.

| Step | Description of Step | Detailed Contents of the Training |
|---|---|---|
| Development learning step | Learning the basic theory on VR contents using a game engine | - Introduction of contents development<br>- Coding theory and basic use of game engine (C# programming)<br>- Data structure and algorithm<br>- Learning how to produce contents using a shooting game<br>- Implementation of team project development with simple logic |
| Development application step | Learning 3D contents production for VR contents production | - Physical theory (vector, motion equation) in 3D environment and 3D contents development practice<br>- 3D contents presentation and development practice (IK Animation, Particle System, etc.)<br>- Understanding the development framework focusing on practice (State Machine, Profiler, etc.)<br>- Fulfilment of 3D contents team project |
| VR Development application step | Understanding VR contents and Learning production skills | - VR technology theory and VR contents development theory<br>- Theory and method of using VR devices (HMD, Simulator, Controller)<br>- Presentation method of VR contents (reduction of cognitive dissonance, increase of level of flow)<br>- VR-based contents production project |
| Industrial project fulfilment step | Fulfilment project together with cooperating company | - Support of necessary technical training specifically for project fulfilment for each team<br>- Checklist to evaluate the detailed implementation for the project<br>- Demonstration and presentation. Job offering and hiring |

In the development learning step, the training was conducted to deliver basic knowledge on programming and necessary theory and principles for developing contents since 70% of trainees did not have any software related background. The trainees were required to practice with the applied problems using their understanding of principles and theories learnt from the training course to improve their problem-solving skills.

The trainees will also be required to carry out a project with simple logic. The project was designed to be carried out with the training contents as shown in Table 4 by means of basic functionalities supported by a development engine. The completed outcome of the project is depicted in Figure 4. The figure to the right of Figure 4 shows contents that allows a player to roll a spherical ball with different weights in a given environment using the basic principles of physics. The objective of this content is to roll the ball to reach the finish line. The trainees could master the basic capability necessary for implementing VR contents such as basic programming and contents development theory after completing this project.

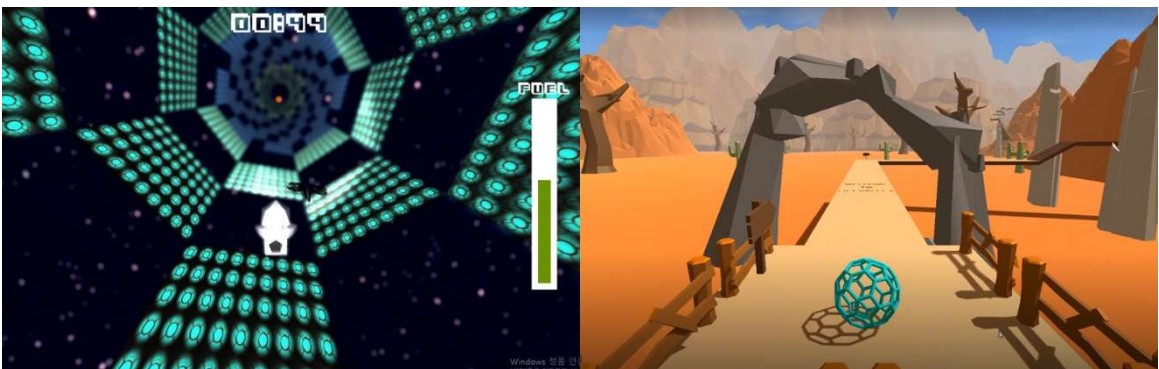

**Figure 4.** Development learning step—Beta Project.

In the development application step, the training was conducted focusing on the understanding of contents representation in a 3D environment. This training phase was designed to learn the advanced techniques such as applied skills to enrich the acquired knowledge from the previous phase. The training contents consist of the development frameworks used in the industrial field and the theory of physics to build a 3D environment.

The project on this phase includes creating a third person viewpoint contents using the training contents shown in Table 4. The final project outcome on this phase is shown in Figure 5. The training was designed to allow trainees to learn the way of representation of contents in a 3D environment and have a deep understanding of advanced scripting.

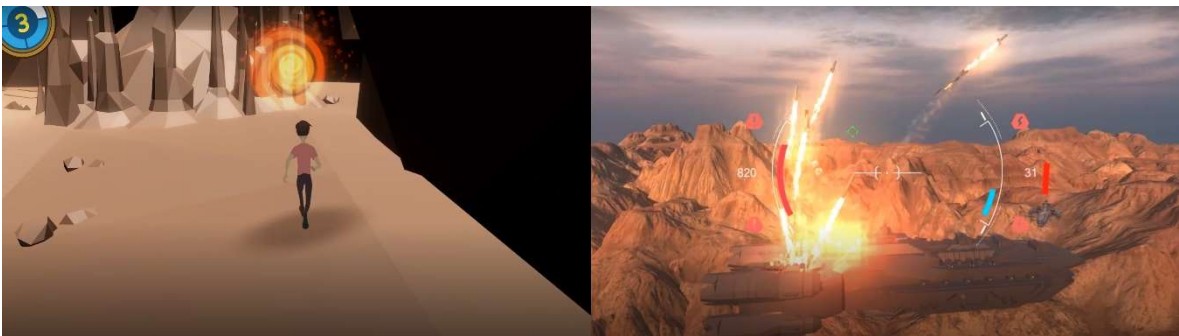

**Figure 5.** Development application step—Beta Project.

In the VR development application step, the trainees were expected to learn development skills for VR contents production and utilizing HMDs and VR devices by making full use of the basic

and advanced techniques acquired from the previous trainings. The ways of reducing the effects of cognitive dissonance and enhancing the level of flow were applied and feedback from the industrial party were reflected. Figure 6 shows the final project outcome on this phase. The trainees are expected to be qualified enough to carry out an industrial project by learning VR contents production and usage of VR device's SDK and API after completing this phase of the training.

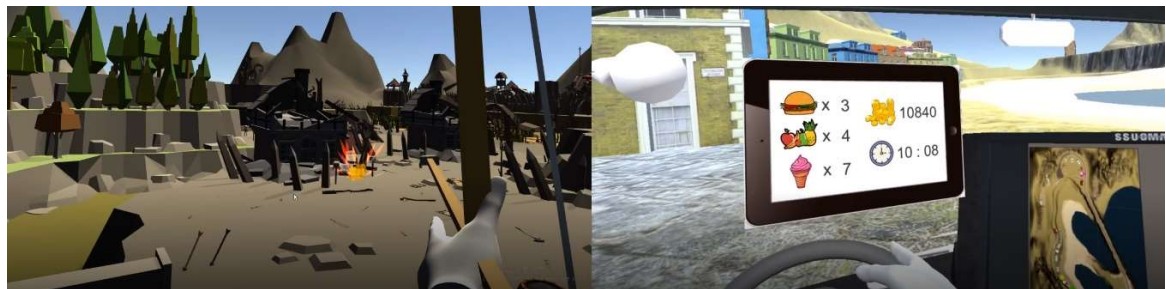

**Figure 6.** VR development application step—Beta Project.

In the industrial project fulfilment step, the qualified trainees from basic and advanced training were expected to be ready for work on an industrial project. The project requirements were requested from the cooperating companies. Introductory sessions for presenting the overview of cooperating company and project details will be held as soon as the project requirements are received.

The details about the projects as proposed by several cooperating companies are presented in Table 5. These projects will be assigned to the trainees and weekly meetings will be held with their mentors to monitor the progress of the project fulfilment. The weekly meetings were critically important opportunity in which mentors could impart valuable knowledge and skills to the trainees to enhance the understanding of the actual work in practical sense. From an industrial party's perspective, it can be an important chance to ensure the enrichment of trainees' skills and qualifications which can positively influence the hiring of trainees.

**Table 5.** Industrial project fulfilment step—Project List.

| Name of Company | Description of Project |
|---|---|
| Visual Dart [31] | "Biology education project using VR"—represent dinosaur's biological characteristics using VR. |
| NextepStudio [32] | "Bloodcode multi"—Convert single-player bloodcode game using VR to multiplayer game & Upload the game to STEAM |
| Sigong Media [33] | "Educational VR content production"—Represent structure of cancer cell and DNA using VR, expression of cancer cell. Create an educational content explaining immunization system and DNA withing human body |
| SESISOFT [34] | "Subculture VR community development and operation"—Support community and build information site for subculture users in VR environment |
| Tekville [35] | "Safety experience content development"—Develop safety experiencing content using HMD for PC and mobile users |
| Technoblood [36] | "VR Hit-Homerun minigame"—Based on users data of VR Hit-Homerun game, everyday new minigame. |

The presentation of the final outcome will be held after completion of the industrial project. External specialists and mentors from the cooperating companies will be invited to assist on the demonstration and evaluations of the project outcomes. Figure 7 shows the outcome of the carried out industrial projects.

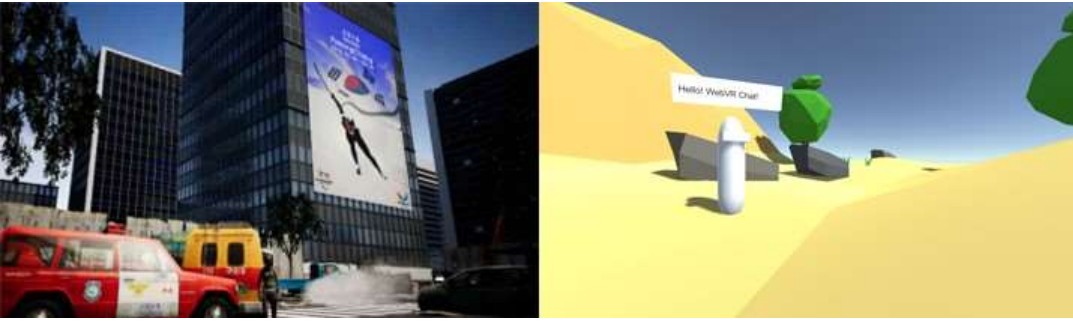

**Figure 7.** Industrial project fulfilment step—Beta Project.

### 4.4. Satisfaction Survey for VR Contents Training Participants

An overall training satisfaction survey was conducted for training participants in order to evaluate the effectiveness of training design. The survey results are presented in Table 6. The trainee's satisfaction was measured using 5-point Likert scale [37] statements ranging from very satisfied (5), satisfied (4), neutral (3), dissatisfied (2) to very dissatisfied (1).

**Table 6.** Satisfaction results

| Category | Question | 5pt * | 4pt | 3pt | 2pt | 1pt | Avg Point |
|---|---|---|---|---|---|---|---|
| Training Contents | Are the training contents beneficial to you? | 18 | 5 | 2 | 1 | 0 | 4.54 |
| | Are the training operated appropriately in according to your intention to participate? | 19 | 5 | 2 | 0 | 0 | 4.65 |
| | Are the training hours and duration appropriate? | 12 | 3 | 5 | 5 | 1 | 3.77 |
| | Are the training materials appropriate? | 19 | 4 | 3 | 0 | 0 | 4.62 |
| Lecturer | Are the ways of teaching appropriate? | 19 | 6 | 1 | 0 | 0 | 4.69 |
| | Do you think that the lecture includes specialized techniques and knowledge? | 22 | 4 | 0 | 0 | 0 | 4.85 |
| Education Environment | Are you satisfied with the overall condition of training facilities? | 15 | 3 | 5 | 3 | 0 | 4.15 |
| | Are enough equipment and materials ready for training? | 19 | 5 | 2 | 0 | 0 | 4.65 |
| Project training | Is difficulty of project training appropriate for you? | 13 | 5 | 6 | 2 | 0 | 4.12 |
| | Are you confident in doing fieldworks once you finished the project training? | 15 | 2 | 7 | 2 | 0 | 4.15 |
| | Do you want to get a job in project-related company? | 12 | 4 | 10 | 0 | 0 | 4.08 |
| Training Satisfaction | Are you satisfied with the program taking into account the training contents, lecturer, training environment, and support? | 16 | 8 | 2 | 0 | 0 | 4.54 |
| | Do you have an intention to recommend this training to others? | 19 | 3 | 4 | 0 | 0 | 4.58 |

\* pt = Point.

The result shows that the average score of satisfaction was 4.4 of 5which indicates that the trainees are satisfied with the overall training. Specifically, the question "are the ways of teaching?" marks the highest score of 4.7. It indicates that the way of training and its contents were appropriate. On the other hand, the question "are the training hours and duration appropriate?" marks the lowest score of 3.8. This indicates that the trainees feel that training hours and duration were not enough. Thus, this suggests that it is necessary to review those factors.

## 5. Conclusions

Virtual reality (VR) technology is rapidly evolving as its industry is continuously growing [38–40]. As the size of VR industry is increasing, the limited number of qualified VR personnel remains as an issue. Thus, this study proposed an educational model aiming to enrich human resources to meet the industry's requirements. In addition, training courses based on the proposed models were launched and implemented.

Educational institutions must enter into mutual agreements with firms in VR industries prior to the start of training. A training curriculum was then designed and implemented on top of the proposed models. The project-based learning approach was incorporated to emphasize the improvement of problem-solving skills for the trainees. The trainees were required to carry out the industrial project proposed by the counter-part VR company where mentoring will be initiated to monitor the progress of the project. In this regard, the trainees can have a profound understanding on how VR contents were developed in the actual industry. Moreover, the trainees can have the opportunity to acquire development knowledge and skills while communicating with their corresponding mentors.

Considering the purpose of this study to address the issue of limited qualified workers, there were achievements in that four trainees were hired by cooperating companies, three were hired by other companies, and four started their own business. Based on these achievements and analysis results, this study agrees with the findings of previous studies dealing with industrial requirements-based education wherein a close relationship between the industrial party and educational institution has a significant importance [30,41,42].

The results of this study indicate that a cooperative relationship between an educational party and industrial party is critical in effectively training and enriching VR contents developers. The findings were expected to be utilized in fostering human resources satisfying the industrial requirements.

However, the effectiveness of the proposed model was limited because the analysis was only done using cases but the variables that might be changed through training were not measured. Thus, in the future, it will be necessary to set the trainee's development capability as a variable, then measure and compare the values before and after the training.

**Author Contributions:** Data curation, H.K.; Project administration, J.K.; Resources, H.K.; Supervision, J.K.; Writing—original draft, H.K.; Writing—review and editing, J.K. All authors have read and agreed to the published version of the manuscript.

**Funding:** This work was supported by Institute of Information & communications Technology Planning & Evaluation (IITP) grant funded by the Korea government(MSIT) (No. 2018-0-01752, Course in SW-based Industry Convergence for VR/AR Developer).

**Acknowledgments:** This work was supported by the National Research Foundation of Korea (NRF) grant funded by the Korea government(MSIT) (No. 2019R1G1A110034111).

**Conflicts of Interest:** The authors declare no conflict of interest.

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
