# Peer review of "A Study on Design and Case Analysis of Virtual Reality Contents Developer Training based on Industrial Requirements"

_electronics, doi:10.3390/electronics9030437_

Round 1
Reviewer 1 Report
This article proposes a study on design and case analysis of VR contents development course, which is built up the cooperative projects and mentoring by institutions and industrial companies. Table 2 provides educational institutions and courses related to VR content, however, all of these educational institutions seem to be in Korea. Could you provide global educational institutions and courses related to VR content?
The quilty of fig. 3 should be improved. However, the main problem of this article is that the topic is suitable to submit to this journal? I think that the education sciences may be suitable to publish this study.
Author Response
Thank you for your opinion.
we reply you the answer below, reflected your opinion.
Response to Reviewer Comments
Point 1: Table 2 provides educational institutions and courses related to VR content, however, all of these educational institutions seem to be in Korea. Could you provide global educational institutions and courses related to VR content?
Response: Table 2 is a study of VR Contents Developer Training curriculum in Korea. Later, we will expand the scope of our research globally in future works
Point 2: The quality of fig. 3 should be improved.
Response: We have attached of the original file

Reviewer 2 Report
In this manuscript, the authors present a new design of training courses aiming to supply specialized workers for virtual reality (VR) content production. Their method employs the problem-based learning paradigm and requires a cooperative relationship between educational institutions and industrial partners. The authors designed, launched, and evaluated training courses based on their method. The proposed approach seems to be reasonable, especially in the industry 4.0 context. The outcomes resulting from the experiment carried out are positive and convincing. Your recommendations are worth implementing. I have only a few minor comments and questions:
1. Could you briefly summarize in what your proposition is better than other curses carried out by educational institutions enumerated in table 2?
2. What was the number of trainees, 27 or 28 (row 298 vs. Table 3)?
3. Table 6 summarizes the trainees' satisfaction. Is it possible to obtain similar statistics when it comes to industrial partners involved in this experiment?
Author Response
Thank you for your opinion.
we reply you the answer below, reflected your opinion.
Response to Reviewer Comments
Point 1: Could you briefly summarize in what your proposition is better than other courses carried out by educational institutions enumerated in table 2?
Response: We add a difference about the other courses to the paper(page 4, line 80~83)
Point 2: What was the number of trainees, 27 or 28 (row 298 vs. Table 3)?
Response: We modified it to 27 trainees. (page 9, line 302)
Point 3: Table 6 summarizes the trainees' satisfaction. Is it possible to obtain similar statistics when it comes to industrial partners involved in this experiment?
Response: The industrial partners serves as a mentor for the trainees. It is difficult to be an similar statistics in this experiment

Round 2
Reviewer 1 Report
I accept the authors' responses. Hence, I accept this revised paper for publication.